# Altered Cerebral Vasoreactivity on Transcranial Color-Coded Sonography Related to Akinetic-Rigid Phenotype of Parkinson’s Disease: Interim Analysis of a Cross-Sectional Study

**DOI:** 10.3390/brainsci13050709

**Published:** 2023-04-24

**Authors:** Rodrigo Tavares Brisson, Rita de Cássia Leite Fernandes, Josevânia Fulgêncio de Lima Arruda, Thiffanny Cristini Cassiano da S. M. Rocha, Nathália de Góes Duarte Santos, Liene Duarte Silva, Marco Antônio Sales Dantas de Lima, Ana Lucia Zuma de Rosso

**Affiliations:** Department of Neurology, Hospital Universitário Clementino Fraga Filho, Universidade Federal do Rio de Janeiro (UFRJ), Rio de Janeiro 21941-617, Brazil

**Keywords:** Parkinson’s disease, cerebrovascular disease, cerebral vasoreactivity, ultrasound application, cerebral hemodynamics

## Abstract

Background: A correlation between worse functional outcomes in Parkinson’s disease (PD) patients with cerebrovascular disease (CVD) or the Akinetic-rigid phenotype has been argued in recent studies. We aimed to evaluate the association of cerebral hemodynamics impairments, assessed by Transcranial Color-coded Doppler sonography (TCCS), on PD patients with different phenotypes of the disease and with risk factors for CVD. Methodology: Idiopathic PD patients (n = 51) were divided into motor subtypes: Akinetic-rigid (AR) (n = 27) and Tremor-dominant (TD) (n = 24) and into two groups regarding vascular risk factors: when ≥2 were present (PDvasc) (n = 18) and <2 (PDnvasc) (n = 33). In a parallel analysis, the Fazekas scale on brain magnetic resonance imaging (MRI) was applied to a sample to assess the degree of leukoaraiosis. TCCS examinations were prospectively performed obtaining middle cerebral artery Mean Flow Velocities (Vm), Resistance Index (RI), and Pulsatility Index (PI). The Breath-Holding Index (BHI) was calculated to assess cerebrovascular reactivity (cVR). Standardized functional scales were administered (UPDRS III and Hoehn&Yahr). Results: The phenotype groups were similar in age, disease duration and demographic parameters, but there were significantly higher H&Y scores than TD group. cVR was impaired in 66.7% of AR vs. 37.5% of TD. AR group exhibited lower BHI (0.53 ± 0.31 vs. 0.91 ± 0.62; *p* = 0.000), lower Vm after apnea (44.3 ± 9.0 cm/s vs. 53.4 ± 11.4 cm/s; *p* = 0.003), higher PI (0.91 ± 0.26 vs. 0.76 ± 0.12; *p* = 0.000) and RI (0.58 ± 0.11 vs. 0.52 ± 0.06; *p* = 0.021). PDvasc group showed higher PI (0.98 vs. 0.76; *p* = 0.001) and higher frequency of altered cVR (72.2% vs. 42.2%; *p* = 0.004). There was a significant predominance of higher values on Fazekas scale in the PDvasc group. We found no difference between the Fazekas scale when comparing motor subtypes groups but there was a trend toward higher scores in the AR phenotype. Conclusions: TCCS, a cost-effective method, displayed impaired cVR in Parkinsonian patients with risk factors for CVD with higher degree of MRI leukoaraiosis. PD patients with the AR disease phenotype also presented impaired cVR on TCCS and greater functional impairment, although with just a trend to higher scores on MRI Fazekas.

## 1. Introduction

Parkinson’s disease (PD) is a neurodegenerative disease characterized by loss of dopaminergic neurons of the mesencephalic substantia nigra. Intracellular and neuritic Lewy body inclusions, composed mainly by abnormal alfa-synuclein protein, are the hallmark of this disease [1]. Even though described two centuries ago and still intensively investigated, many physiopathological aspects of PD remain to be elucidated. In addition, to date, the disease has no definite biological marker identified, being diagnosed on clinical grounds [2]. One of the diagnostic exclusion criteria for PD is a positive stroke history temporally related to the Parkinsonian syndrome. However, vascular lesions are common incidental findings in pathologically confirmed PD [3,4]. Furthermore, it remains unknown, or the evidence is scarce, whether is there an additive effect between the neurodegenerative loss of neurons and the burden of small vessel disease (SVD) in idiopathic PD [5,6,7,8]. Some papers have indeed found worse outcomes in PD individuals with risk factors for cerebrovascular disease (CVD), as evidenced by a higher degree of disability and premature death [9,10,11,12].

Some studies, as well, have discussed the role of the so-called Neurovascular Unit (NVU) in the neurodegenerative process in different diseases such as Alzheimer’s disease, multiple sclerosis and PD [13,14]. Structurally, NVU is composed of the following basic elements: endothelial wall cells, glial cells and perivascular nerve endings [15,16]. Thus, processes that are involved in the dysfunction of any NVU elements affect the normal control of the physiological mechanisms of cerebral blood flow (CBF) control, which can lead to injury to the cerebral nervous tissue [15,16,17]. The NVU dysfunction is linked to cerebral vasoreactivity (cVR) dysfunction which, in turn, has been associated with worse clinical presentation in some PD patients [17,18,19]. CBF is regulated by oxygen (O_2_) and carbon dioxide (CO_2_) tensions in blood so that increased CO_2_ stimulates an increment in CBF, while an increase in O_2_ decreases it. This process of increasing or decreasing CBF is carried out by interactive mechanisms between the cells of the NVU that result in contraction or relaxation of small-caliber vessels, the so-called cVR [17]. There are two methods for cVR evaluation: through measurement of the response of blood velocities in cerebral vessels acquired by transcranial ultrasound after vasodilation stimuli (hypercapnia or the use of intravenous acetazolamide) or by detecting a local increase in the extraction fraction of O_2_ by positron emission tomography (PET) [17,20].

MRI is not an essential tool for the diagnosis of idiopathic PD since it is unremarkable in these cases, but it is important in the search for evidence of atypical Parkinsonism or other secondary Parkinsonian syndromes [21]. However, the role of MRI in the investigation of silent white matter lesions (WML) in individuals with idiopathic PD has been recently more intensively discussed, as in a study conducted by Bohnen et al., 2011 that found an important WML burden in elderly individuals with PD that may account for some motor and cognitive deficits in these individuals, contributing to a worse outcome [4]. In this context, it is also known that WML are associated with comorbidities such as hypertension and diabetes, which, in turn, are associated with cerebral vascular dysfunction through a higher degree of endothelial cells lesions, causing changes in cVR. Some studies revealed that patients with PD have altered cVR compared to healthy individuals [22,23].

PD patients may be classified into three different clinical phenotypes: one in which tremor is the predominant manifestation (Tremor-dominant type), another in which the dominant symptom is rigidity (Akinetic-rigid type) and a third phenotype with no predominant motor manifestation (Mixed type) [24]. A previous small study of our research group evidenced an association between these clinical phenotypes of PD and the presence of risk factors for CVD associated with altered cVR found in transcranial Doppler [5]. However, despite this evidence, none of the previous studies correlated MRI findings, cVR sonographic data, PD motor phenotype, or presence of risk factors for CVD in PD patients together.

All these data prompt our research group to investigate the vascular component of NVU through the study of cVR with Transcranial Color-coded Doppler sonography (TCCS). Furthermore, we aimed to investigate the degree of cerebral WML through MRI and to associate these data with the phenotypic expression of PD patients.

So, our objectives were: (1) to check out for an association between different motor PD phenotypes with the presence of known risk factors for CVD and with the hemodynamic changes depicted by TCCS, and (2) to investigate an association between altered cerebral hemodynamics, evaluated by TCCS, a real-time, cost-effective and non-invasive method, with WML burden on MRI.

## 2. Materials and Methods

The study was conducted in a tertiary Neurology outpatient clinic, specialized in Movement Disorders. Subjects were recruited randomly from August 2020 through August 2022. We included PD patients who were ≥50 years old at disease onset and had a diagnostic length of ≥2 years, made by a neurologist specialized in movement disorders. All of the patients fulfilled the Movement Disorders Society (MDS) Clinical Diagnostic Criteria for Parkinson Disease [2] and signed the informed consent form. Exclusion criteria were a history of intracranial surgery, traumatic brain injury, stroke, carotid disease with hemodynamic repercussions or an inadequate transtemporal window for the TCCS examination, which is an inability to detect sonographic signals from the middle cerebral artery, mainly due to temporal increased bone thickness or porosity [25]. Those patients unable to hold their breath during the apnea test for at least 15 s were also excluded. The study protocol was approved by the local ethics committee, number CAAE: 45064321.0.0000.5257, and all patients provided written informed consent for the procedure.

To answer the first objective, PD patients were divided into two groups according to clinical signs comprising motor phenotypes: Akinetic-rigid (AR) or Tremor-dominant (TD). We used a formula to classify patients phenotypes derived from the one proposed by Kang and coworkers [24] using UPDRS part III (Unified Parkinson’s Disease Rating Scale). Our modified formula divided a Tremor score (T)—that is the sum of UPDRS-III score of a patient in items 20 and 21 divided by 4, by the Akinetic/rigid score (A)—that is the sum of UPDRS-III score of the patient in items 22 to 27 and 31 divided by 15. The formula is as follows: Phenotype (Phen) = T/A = (i20 + i21/4)/(i22 + i23 + i24 + i25 + i26 + i27 + i31/15). PD patients scoring Phen ≤ 1.0 were classified as AR phenotype while those patients scoring Phen > 1.0 were classified as TD phenotype.

The patients were further classified into two different groups comprising PD patients with two or more CVD risk factors (hypertension, diabetes, dyslipidemia, heart disease, lung disease, kidney disease, obesity, or smoking)—the PDvasc group. The other group included PD patients with no more than one CVD risk factor—the PDnvasc group as established by Malek et al., 2016 [18].

The assessment of PD patients regarding disease severity was carried out using the modified scale of Hoehn&Yahr (H&Y) [26] and the Unified Parkinson’s disease rating scale (UPDRS) part III. Cognitive status was assessed by performing the Montreal Cognitive Assessment Test (MoCA) [27], and the clinical profiles were evaluated using a structured questionnaire.

To investigate cerebral hemodynamics, all of the patients underwent TCCS in the Neurosonology laboratory with a HD11XE (Philips Healthcare, Eindhoven, The Netherlands) using a sectorial transducer of 1–4 MHz. Examination of intracranial vascularity followed established protocols [25] and provided data on the mean blood flow velocity (Vm), pulsatility index (PI), and resistance index (RI) of the middle cerebral artery (MCA) in a resting state (Figure 1A). The values obtained from one temporal window were used for analysis (usually from the right side). Cerebrovascular reactivity (cVR) was measured using the breath-holding index (BHI) as described by Markus and Harrison [28]. The BHI is obtained by measuring mean flow velocities in MCA while the patient is at rest (Vmr) and then when holding the breath for a minimum of 15 s, immediately after restart of breathing (Vma) (Figure 1B). The BHI is calculated according to the following formula: BHI = (Vma − Vmr)/Vmr × 100/t’ where Vma is the mean MCA blood velocity after apnea, Vmr is the mean MCA blood velocity speed at rest, and t’ is apnea duration in seconds. BHI values of <0.69 were considered abnormal (impaired cVR) [28].

To answer the second objective, brain MRI was performed using a Siemens Avontode 1.5 T (Erlangen, Germany) equipment in a group of patients of the sample (n = 17). To quantify the burden of white matter injury (or WML), the modified Fazekas scale [29] was applied in MRI FLAIR images (Fluid-attenuated inversion recovery) (Figure 2).

The statistical program SAS Institute Inc. 2016, SAS/ACCESS 9.4, was used for data analysis, computed as the mean ± standard deviation (SD) for quantitative variables or frequency (%) for qualitative variables. The normal distribution of data allowed the use of parametric tests. To investigate a difference in the distributions of quantitative variables between the groups, the t test was performed for independent samples. For qualitative variables, the Chi square test was used. Statistical significance was set at *p* < 0.05.

## 3. Results

Of the 66 patients who were invited, 10 were excluded due to transtemporal window failure after TCCS examination, three because they were unable to perform the apnea test for more than 15 s and two because they were using a deep-brain stimulation device (DBS). Fifty-one PD patients were included in this study. There was a white male predominance, with a mean age of 65.9 ± 8.3 years (Table 1). Most individuals had a schooling history longer than 8 years (62.8%). The mean disease duration was 6.12 ± 3.4 years, and 78.5% of patients had a disease duration of <10 years. The mean H&Y scale score was 2.36 (mild bilateral disease with recovery on pull test), and the mean UPDRS part III scale score was 19 points. The analysis of cognitive status using MoCA revealed a mean in the entire cohort of 19.6 out of a maximum of 30 points. All the patients were basically treated with levodopa with a mean daily dose of 542.9 mg/day.


**Analysis of the sample according to motor phenotype**


Considering the motor subtypes of PD, 27 patients were allocated in the AR group and 24 in the TD group. As shown in Table 1, regarding disease severity, the AR subtype showed higher H&Y grading than the TD subtype (2.7 vs. 1.9; *p* = 0.024), but similar UPDRS III scoring. The AR group was not significantly older than TD group 67.8 ± 8.18 y vs. 63.7 ± 8.08 y (*p* = 0.605), had similar disease duration, were using similar levodopa doses and scored also similarly in the cognitive test MoCA. Nonetheless, AR subtype patients showed higher RI (0.58 ± 0.11 vs. 0.52 ± 0.06; *p* = 0.021), higher PI (0.91 ± 0.26 vs. 0.76 ± 0.12; *p* = 0.000), lower BHI (0.53 ± 0.32 vs. 0.91 ± 0.62; *p* = 0.00) and lower Vma (44.3 ± 9.0 cm/s vs. 53.4 ± 11.4 cm/s; *p* = 0.003), leading to more altered cVR.

Regarding classification of WML burden on MRI, PD patients (n = 51) ranged from 0 to 2 on Fazekas’ scale, with 82.3% falling on grades 0 or 1. There was no difference in scoring between the phenotypes (Table 1).


**Analysis of the sample according to vascular risk factors**


Eighteen patients entered the PDvasc group and 33 the PDnvasc group, according to the presence of ≥2 vascular risk factors; the most frequent were hypertension (66.7%), smoking (31.4%), and heart disease (25.6%), followed by diabetes (DM2) (11.1%) (Table 2). Demographic and clinical characteristics of these PD groups are shown in Table 3. They had overall similar characteristics regarding all demographic parameters as well as similar disease duration and severity of symptoms. Vascular analysis, however, showed significant differences between the groups. As expected, PDvasc patients had higher RI (0.59 vs. 0.52; *p*= 0.030), higher PI (0.98 vs. 0.76; *p* = 0.001) and higher frequency of altered cVR (72.2% vs. 42.2%; *p* = 0.004). PDnvasc showed higher Vma values: 50.9 ± 0.57 cm/s vs. 44.2 ± 0.33 cm/s (*p* = 0.028). Albeit lower, BHI mean values of the PDvasc group failed to reach significance (0.55 vs. 0.80; *p* = 0.144).

MRI-based Fazekas’ score showed PDvasc group, as expected, ranging predominantly in higher grades 1 or 2 (80%), while in the PDnvasc group, 41.7% of patients fell in grade 0 and 58.3% in grade 1.

Regarding PD phenotype, AR patients were equally distributed in the vascular groups. However, there was a predominance of the TD phenotype in PDnvasc patients (75%) (Table 3).

## 4. Discussion

We investigated PD patients with TCCS, a real-time, cost-effective, and non-invasive method used in the evaluation of cerebral hemodynamics. Our results revealed that 66.7% of patients with a predominantly rigid phenotype (AR group) have cVR impairment. In contrast, patients with the tremulous motor phenotype (TD group) seem to exhibit impaired cVR less often, in 37.5% (Table 1). It is important to note that these two groups of patients were similar in age, sex and disease duration, differing only in the severity of symptoms at the moment of the exam—the AR group scoring higher on H&Y scale.

As recent works suggest, the rate of progression seems to be slower and the severity of functional disability seems to be higher in PD individuals with AR phenotype when compared to those with TD phenotype [30,31]. These differences may originate in different neural pathways derangements: mesolimbic-cortical pathway in AR and cerebellar-thalamic projections in TD phenotype [30,32]. Whether vascular compromise also plays a role in Parkinsonians outcomes remains a matter of debate.

It is already known that small-vessel disease leads to a dysfunction of the NVU. Consequently, cVR is impacted and may result in hypoxia, damage to the blood–brain barrier and neuroinflammation [13,19]. This mechanism has been classified as a possible trigger for the worsening of the neurodegenerative process, clinically evidenced by greater functional motor impairment in PD patients or greater cognitive impairment in Alzheimer’s disease, for example. Radiologically, one can expect an increased burden of WML on MRI of more committed patients, which, indeed, has recently been described [4,13,14,33].

Changes in cVR are also associated with diminished vascular functionality in individuals with hypertension in middle-age and may serve as a preclinical marker for brain dysfunction later in life [34]. In addition, individuals with risk factors for CVD may have reduced vascular reactivity across different cortical regions [35,36,37,38,39]. Our results revealed that PD patients with two or more vascular risk factors (PDvasc group) exhibit hemodynamics data indicating brain vascular pathology. These patients had a higher rate of changes in cVR with lower BHI, higher PI and RI on TCCS evaluation. Despite having similar disease duration, the mean H&Y scale score was significantly higher in individuals in the PDvasc group in our sample, suggesting a possible influence of vascular risk factors on the clinical severity of PD. Other published papers found that patients with vascular risk factors are also more prone to rapid deterioration and increased mortality [9,10,18]. In accordance with a previous study, the presence of two or more vascular risk factors is associated with worse motor impairment, as well as cognitive decline [5,18].

Sartor J. et al., 2017 found a relevant association of age and PD with white matter lesions, gait and executive function deficits [40]. In our study, we found similar data when comparing PDvasc × PDnvasc groups. The Fazekas scale showed a higher frequency of patients with higher scores at the PDvasc group. However, despite having a higher frequency of patients with a higher degree of leukoaraiosis (quantified by the Fazekas scale) in the group of patients with the AR phenotype, there was no significance for this analysis. A recent study found that patients with the AR subtype have more iron deposition in the globus pallidus than patients with the TD type [41]. Iron deposition in brain tissue has also been reported to be increased in patients with CVD [42], which could add an explanation to the differences in vascular Doppler parameters found in our study and should be explored in future research, perhaps with functional MRI.

It is our assumption that all these data corroborate with the hypothesis that the neurodegenerative process in PD is influenced by rather clinically silent damage to small vessels, and that patients presenting with risk factors for CVD could have an exacerbation of NVU dysfunction and, consequently, of the cVR. These processes together may accelerate the neurodegenerative process and account for the different clinical phenotypes and outcomes of PD patients.

To the best of our knowledge, this is the first study to investigate either a correlation between hemodynamic data, using TCCS, in PD patients with different motor subtypes of the disease and to compare PD patients without vascular risk factors for CVD. Both analyses took into account WML in brain MRI. The understanding that this hemodynamic alteration may be associated with the neurodegeneration process and that this leads to a worsening of vascular impairment and vice versa (leading to a vicious cycle) may stimulate therapeutic targets aimed at NVU, in addition to the usual treatment of dopamine replacement in the PD central nervous system.

Although limited by the small sample size of this interim analysis, we believe that these results highlight the urgent need for studying the interaction between neurodegeneration and vascular disease in PD.

Due to the high incidence of inadequate transtemporal windows in elderly women, perhaps due to their higher incidence of osteoporosis, the ultrasound study was impossible in most female subjects. This insurmountable fact led to an imbalance in the proportion of men and women in the study up to here. Another limiting factor was the difficulty in MRI acquisition in some PD patients due to involuntary movements during the exam. Some patients were unable to perform the MRI due to the presence of ferromagnetic material in the body and others due to claustrophobia. Despite these difficulties, our results encourage us and we expect other researchers to replicate this work in a prospective cohort with larger number of patients to obtain a better understanding of the burden of CVD in PD patients, and to understand the differences between their motor phenotype presentations and the possibly different prognosis they carry.

## 5. Conclusions

We conclude that Parkinsonian patients with the AR subtype of the disease present altered cVR in TCCS and greater functional impairment than the TD subtype. Although we could not find a difference in Fazekas scale among motor groups, there was a trend toward higher scores in the AR phenotype. In addition, patients with two or more risk factors for CVD present altered cVR in TCCS correlating to higher degree of leukoaraiosis in MRI.

We highlight that TCCS seems to be a cost-effective way of assessing cerebral hemodynamics in PD patients that could help to monitor the disease and the factors contributing to a worse progression. We hope that this line of work contributes to the understanding of the pathophysiological processes involved in Parkinson’s disease, which, after centuries of its first description, is still not fully understood. Further longitudinal studies correlating these findings with larger samples are needed.

## Figures and Tables

**Figure 1 brainsci-13-00709-f001:**
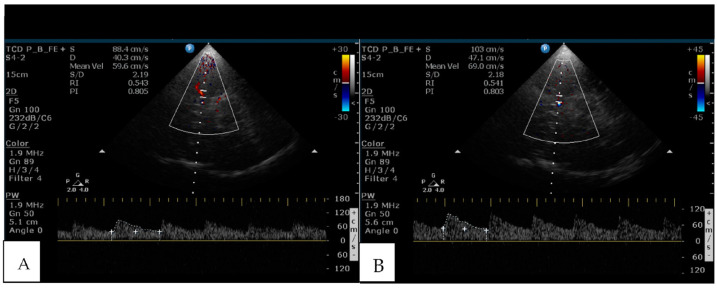
(**A**) Normal Transcranial Color-coded Sonography examination of middle cerebral artery blood mean velocity (Vm = 59.6 cm/s), pulsatility index (PI = 0.805) and resistance index (RI = 0.543) during rest; (**B**) same parameters after 30 s apnea (breath-holding test) evidencing higher velocities in the same patient (Vma = 69.0 cm/s).

**Figure 2 brainsci-13-00709-f002:**
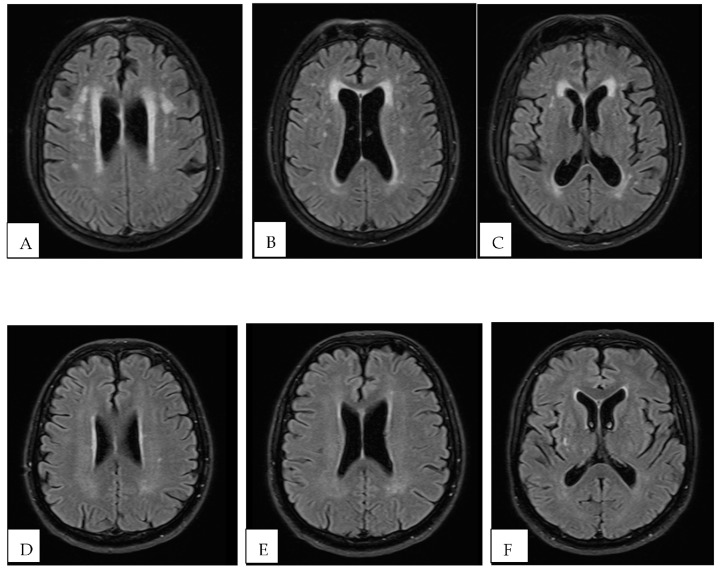
Examples Fazekas scale grading on MRI FLAIR sequence of PD patients. (**A**–**C**) PD patient with a disease duration of 5 years with Akinetic-Rigid phenotype and in the PDvasc group grading 2 on Fazekas scale and altered cVR on TCCS with a breath-holding index (BHI) equal to 0.63 {N > 0.69} (not shown). (**D**–**F**) PD patient with 6 years disease duration with Tremor-Dominant phenotype and in the PDnvasc group grading 1 on Fazekas scale and preserved cVR on TCCS with a BHI equal to 0.95 (not shown).

**Table 1 brainsci-13-00709-t001:** Study groups according to PD phenotypes.

	Total Sample	Akinetic-Rigid	Tremor-Dominant	*p*-Value
**Number of Participants**	**n = 51**	**n = 27**	**n = 24**	
Age (Years—SD)	65.9 ± 8.3	67.8 ± 8.1	63.7 ± 8.0	0.605
Gender n (%) male	42 (82.3)	22 (81.4)	20 (83.3)	0.862
Ethnicity n (%) white	37 (72.5)	23 (85.1)	14 (58.3)	0.032
Schooling n (%) >8 y	32 (62.8)	16 (59.6)	16 (66.7)	0.585
H&Y scale (M ± SD)	2.36 ± 0.95	2.7 ± 1.1	1.98 ± 0.7	0.023
UPDRS-III (M ± SD)	18.88 ± 8.18	19.4 ±8.9	18.3 ± 7.4	0.372
Disease duration y (M ± SD)	6.12 ± 3.4	6.78 ± 3.5	5.38 ± 3.1	0.144
Daily levodopa dose mg (M ± SD)	542.9 ± 25.5	542.6	497.9	0.058
MoCA points in max. 30 (M ± SD)	19.60 ± 6.34	18.9 ± 7.0	21.3 ± 4.9	0.100
RI	0.55 ± 0.09	0.58 ± 0.11	0.52 ± 0.06	0.021
PI	0.84 ± 0.22	0.91 ± 0.26	0.76 ± 0.12	0.005
BHI	0.71 ± 0.51	0.53 ± 0.31	0.91 ± 0.62	0.000
Vma cm/s (M ± SD)	48.6 ± 11.1	44.32 ± 9.0	53.42 ± 11.4	0.003
cVR n (%) *yes = BHI < 0.69*	27 (52.9)	18 (66.7)	9 (37.5)	0.037
Fazekas scale n (%)	17 (100)	8 (47.1)	9 (52.9)	0.127
*0* n (%)	6 (35.3)	2 (25.0)	4 (44.5)	
*1* n (%)	8 (47)	3 (37.5)	5 (55.6)	
*2* n (%)	3 (17.7)	3 (37.5)	0 (0.00)	
*3* n (%)	0 (0.0)	0 (0.00)	0 (0.00)	

BHI: breath-holding index; cVR: cerebrovascular reactivity; PI: pulsatility index; RI: resistance index; Vma: middle cerebral artery mean flow velocities after apnea. H&Y: modified scale of Hoehn&Yahr; Unified Parkinson’s Disease Rating Scale: UPDRS part III; Montreal Cognitive Assessment Test: MoCA.

**Table 2 brainsci-13-00709-t002:** Vascular comorbidities in the sample.

Variable	Total Sample	PDvasc	PDnvasc	*p*-Value
Smoke n (%)	16 (31.4)	7 (38.9)	9 (27.8)	0.730
Type 2 diabetes n (%)	3 (11.1)	6 (33.3)	0 (0)	0.004
Depression n (%)	12 (23.5)	3 (16.7)	9 (27.3)	0.728
Stroke n (%)	0 (0.0)	0 (0)	0 (0)	
HBP n (%)	34 (66.7)	18 (100)	16 (48.5)	0.002
Dyslipidemia n (%)	5 (9.8)	5 (27.78)	0 (0)	0.003
Cardiac disease n (%)	13 (25.6)	12 (66.7)	1 (3.0)	0.001
Lung disease n (%)	4 (7.8)	3 (16.7)	1 (3.03)	0.120
Alcoholism n (%)	11 (21.6)	5 (27.8)	6 (18.1)	0.425

HBP: High blood pressure; PDvasc: Parkinson disease patients with ≥2 vascular risk factors; PDnvasc: Parkinson’s disease patients with <2 vascular risk factors; SD: standard deviation.

**Table 3 brainsci-13-00709-t003:** Study groups according to vascular risk factors.

	Total Sample	PDvasc	PDnvasc	*p*-Value
**Number of Participants**	**n = 51**	**n = 18**	**n = 33**	
Age (Years—SD)	65.9 ± 8.3	70.3 ± 6.7	63.6 ± 8.3	0.0020
Gender n (%) male	41 (80.4)	15 (83.3)	26 (78.8)	0.696
Ethnicity n (%) white	37 (72.5)	14 (77.8)	23(69.7)	0.380
Schooling n (%) >8 y	32 (62.8)	13 (72.2)	19 (57.6)	0.301
H&Y scale (M + SD)	2.36 ± 0.95	17.61 + 4.02	19.57 + 5.37	0.411
UPDRS-III (mean)	18.88 ± 8.18	17.61 ± 4.02	19.57 ± 5.37	0.411
Disease duration y (M + SD)	6.12 ± 3.4	6.11 (4.25)	6.12 (2.11)	0.889
Daily levodopa dose mg (M + SD)	542.9 ± 25.5	614.70 ± 23.5	504.68 ± 27.8	0.317
MoCA points in max. 30 (M + SD)	19.60 ± 6.34	20.35 ± 0.57	19.1 ± 0.46	0.644
RI	0.55 ± 0.09	0.59 ± 0.66	0.52 ± 0.30	0.030
PI	0.84 ± 0.22	0.98 ± 0.74	0.76 ± 0.35	0.001
BHI	0.71 ± 0.51	0.55 ± 0.27	0.80 ± 0.60	0.144
Vma cm/s (M + SD)	47.5 ± 12.7	44.2 ± 0.33	50.9 ± 0.57	0.028
cVR n (%) *yes = BHI < 0.69*	27 (52.9)	13 (72.2)	14 (42.2)	0.004
Fazekas scale n (%)	17 (100)	8 (47.1)	9 (52.9)	0.012
*0* n (%)	6 (35.3)	1 (20)	5 (41.7)	
*1* n (%)	8 (47)	1 (20)	7 (58.3)	
*2* n (%)	3 (17.7)	3 (60)	0 (0.0)	
*3* n (%)	0 (0.0)	0 (0.0)	0 (0.0)	
PD phenotype n (%)				
Akinetic-rigid (AR)	27 (52.9)	12 (44.4)	15 (55.6)	0.147
Tremor-dominant (TD)	24 (47.1)	6 (25)	18 (75)	

BHI: breath-holding index; cVR: cerebrovascular reactivity; PDvasc: Parkinson’s disease patients with ≥2 vascular risk factors; PDnvasc: Parkinson’s disease patients with <2 vascular risk factors; PI: pulsatility index; RI: resistance index; Vma: middle cerebral artery mean flow velocities after apnea.

## Data Availability

Data available on request due to restrictions eg privacy or ethicalThe data presented in this study are available on request from the corresponding author. The data are not publicly available due to ethical issues.

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
