# Peer review of "Altered Cerebral Vasoreactivity on Transcranial Color-Coded Sonography Related to Akinetic-Rigid Phenotype of Parkinson’s Disease: Interim Analysis of a Cross-Sectional Study"

_brainsci, 2023, doi:10.3390/brainsci13050709_

Round 1

Reviewer 1 Report

The manuscript developed by Tavares Brisson et al, aimed to analyze the impact of the altered cerebral 14 hemodynamics, by transcranial Doppler, on PD patients with risk factors for CVD or with 15 different phenotypes of the disease. In general, the work is innovative, however, I have some recommendations and requests.

Major observations

• Introduction

1. Briefly describe background findings through MRI or cVR sonographic studies in PD.

• Methods:

1. Be more specific in your inclusion criteria

2. Define what is considered to be an inadequate transtemporal window

3. Verify the wording of the categorization between the phenotypes. Since it is a calculated radius.

4. Indicate the number of the project approved by the ethics committee

5. It remains to describe and analyze: 1) the type of treatment, as well as its duration for patients with PD; 2) the time of evolution of the disease; 3) the age of the pathology onset

6. I request that the authors use an up-to-date and exact scale such as Modified Fazecas (10.5535/arm.2014.38.5.620)

7. Detail why only 17 patients had MRI.

8. Specify which MRI sequences were performed

•Results:

1. Be more specific on ethnicity

2. The groups are unbalanced from the statistical point of view, it is not valid to compare them.

3. The tables lack significant differences in some variables

• Discussion:

1. Mention the possible link between imaging alterations and VNU dysfunction present in neurodegenerative diseases, including PD (include the following articles: 10.3389/fnins.2020.00334; 10.3390/ijms22042022)

2. Discuss why patients with PDnvasc develop the pathology earlier than PDvasc.

• References:

1. About 70% of the references are before 2016 and have a different format.

Minor observations

1. In some parts of the text, they repeat the meaning of acronyms and do not add them in others.

2. Figure 2 is not described in the text.

3. Check misspellings.

Author Response

Dear Editor-in-Chief, Prof. Dr. Stephen D. Meriney

Department of Neuroscience, University of Pittsburgh, Pittsburgh, PA 15260, USA

We respect and accept the valuable contribution of the reviewers, with their suggestions and requests for correction.  We have made a major revision of the manuscript and these changes will certainly improve the paper and give greater consistency to it.

Each of the reviewers’ questions on the matter of the manuscript are answered below.

The changes made are highlighted in red in this letter (reply to reviewer) and in the revised manuscript (with revisions highlighted). We hope to have met the expectations of the editorial board.

Answers to Reviewer 1

Reviewer query: The manuscript developed by Tavares Brisson et al, aimed to analyze the impact of the altered cerebral 14 hemodynamics, by transcranial Doppler, on PD patients with risk factors for CVD or with 15 different phenotypes of the disease. In general, the work is innovative, however, I have some recommendations and requests.

Reply to reviewer: I am grateful for the considerations of the reviewer and his enriching observations on the article.

Major observations

  • Introduction
  1. Briefly describe background findings through MRI or cVR sonographic studies in PD.

Reply to reviewer: We appreciate the reviewer's suggestion and have added these changes to the text, as noted in lines 54 to 84 marked in red in the text.

  • Methods:
  1. Be more specific in your inclusion criteria.

Reply to reviewer: Thank you for your observation. We added new data regarding the inclusion criteria highlighted in red on lines 108 to 112 in the text.

  1. Define what is considered to be an inadequate transtemporal window

Reply to reviewer: It is the inability to detect sonographic signal from the middle cerebral artery. We mentioned this new information at lines 114 and 115 in the text.

  1. Verify the wording of the categorization between the phenotypes. Since it is a calculated radius.

Reply to reviewer: We rewrite the text as seen in the lines 120 to 132 in the text.

  1. Indicate the number of the project approved by the ethics committee

Reply to reviewer: 45064321.0.0000.5257 – line 118 in the text.

  1. It remains to describe and analyze: 1) the type of treatment, as well as its duration for patients with PD; 2) the time of evolution of the disease; 3) the age of the pathology onset

Reply to reviewer: these data are shown in table 1.

  1. I request that the authors use an up-to-date and exact scale such as Modified Fazecas (10.5535/arm.2014.38.5.620).

Reply to reviewer: We reanalyzed and rewrote the text as noted in lines 163 and 164.

  1. Detail why only 17 patients had MRI.

Reply to reviewer: Among the limiting factors of the study, described between lines 335 to 341, one of the limiting factors was the difficulty in MRI acquisition in some PD patients due to involuntary movements during the exam. Other patients were unable to perform the MRI due to the presence of ferromagnetic material in the body and other, due to claustrophobia.

  1. Specify which MRI sequences were performed

Reply to reviewer: Modified Fazekas scale was applied using the MRI FLAIR (Fluid-attenuated inversion recovery) sequence to quantify the degree of white matter injury in a group of patients of the sample (figure 2), as shown in line 163 to 165 in the text.

  • Results:
  1. Be more specific on ethnicity

Reply to reviewer: white and non white people

  1. The groups are unbalanced from the statistical point of view, it is not valid to compare them.

Reply to reviewer: We redid the calculations according to the modification of the formula (line 127) and dichotomizing the individuals as TD and RA. This new analysis is discussed in the text in lines 198 to 206 and in table 4.

  1. The tables lack significant differences in some variables

Reply to reviewer: Please consider the discussion above

  • Discussion:
  1. Mention the possible link between imaging alterations and VNU dysfunction present in neurodegenerative diseases, including PD (include the following articles: 10.3389/fnins.2020.00334; 10.3390/ijms22042022)

Reply to reviewer: We greatly appreciate the suggestion and have included this topic throughout the text (item "discussion") highlighted in red.

  1. Discuss why patients with PDnvasc develop the pathology earlier than PDvasc.

Reply to reviewer: Their disease duration is the same.

  • References:
  1. About 70% of the references are before 2016 and have a different format.

Reply to reviewer : Thank you for your observation and as per your suggestion, we have included some more up-to-date references.

Minor observations

  1. In some parts of the text, they repeat the meaning of acronyms and do not add them in others.

Reply to reviewer: We note the errors and correct them throughout the text.

  1. Figure 2 is not described in the text.

Reply to reviewer: We have corrected this data and included the reference to figure 2 in the line165.

  1. Check misspellings.

Reply to reviewer: We note the errors and correct them throughout the text

We greatly appreciate the attention given to the analysis and contribution to the enrichment of this article.

Sincerely,

The authors

Reviewer 2 Report

The vascular context of Parkinson's disease is not fully recognized, however in this work:

1. The number of examined patients is relatively low, moreover the number of male and female is disproportional, which should be considered as a limitations.

2. Authors should elaborate more widely on vascular risk factors and their possible impact on pathogenesis of parkinsonisms not necessarily Parkinson's Disease e.g. atypical parkinsonisms - Ref.

Could hyperlipidemia be a risk factor for corticobasal syndrome? - a pilot study. Neurol Neurochir Pol. 2022 Dec 15. doi: 10.5603/PJNNS.a2022.0078. Epub ahead of print. PMID: 36519660.

Inflammation-associated peripheral blood cells and serum lipid levels are associated with Parkinson's disease. Chin Med J (Engl). 2022 Nov 7. doi: 10.1097/CM9.0000000000002397. Epub ahead of print. PMID: 36921109.

The Rate of Decrease in Brain Perfusion in Progressive Supranuclear Palsy and Corticobasal Syndrome May Be Impacted by Glycemic Variability-A Pilot Study. Front Neurol. 2021 Nov 8;12:767480. doi: 10.3389/fneur.2021.767480. PMID: 34819913; PMCID: PMC8606811.   3. It would be valuable to elaborate on future perspectives. 4. Authors should elaborate on links between vascular and inflammatory factors in the neurodegeneration.   Copy

  Copy

Author Response

Dear Editor-in-Chief, Prof. Dr. Stephen D. Meriney

Department of Neuroscience, University of Pittsburgh, Pittsburgh, PA 15260, USA

We respect and accept the valuable contribution of the reviewers, with their suggestions and requests for correction.  We have made a major revision of the manuscript and these changes will certainly improve the paper and give greater consistency to it.

Each of the reviewers’ questions on the matter of the manuscript are answered below.

The changes made are highlighted in red in this letter (reply to reviewer) and in the revised manuscript (with revisions highlighted). We hope to have met the expectations of the editorial board.

Answers to Reviewer 1

Comments and Suggestions for Authors

The vascular context of Parkinson's disease is not fully recognized, however in this work:

  1. The number of examined patients is relatively low, moreover the number of male and female is disproportional, which should be considered as a limitations.

Reply to reviewer: We appreciate the observation, the justification for this limitation is in the description of the limitations of the study between lines 332-341, highlighted in red in the text. Due to the high incidence of inadequate transtemporal windows in elderly women, perhaps due to their higher incidence of osteoporosis, the ultrasound study was impossible in most female subjects. This insurmountable fact led to an imbalance in the proportion of men and women in the study up to here.

  1. Authors should elaborate more widely on vascular risk factors and their possible impact on pathogenesis of parkinsonisms not necessarily Parkinson's Disease e.g. atypical parkinsonisms - Ref.

Could hyperlipidemia be a risk factor for corticobasal syndrome? - a pilot study. Neurol Neurochir Pol. 2022 Dec 15. doi: 10.5603/PJNNS.a2022.0078. Epub ahead of print. PMID: 36519660.

Inflammation-associated peripheral blood cells and serum lipid levels are associated with Parkinson's disease. Chin Med J (Engl). 2022 Nov 7. doi: 10.1097/CM9.0000000000002397. Epub ahead of print. PMID: 36921109.

The Rate of Decrease in Brain Perfusion in Progressive Supranuclear Palsy and Corticobasal Syndrome May Be Impacted by Glycemic Variability-A Pilot Study. Front Neurol. 2021 Nov 8;12:767480. doi: 10.3389/fneur.2021.767480. PMID: 34819913; PMCID: PMC8606811.   3. It would be valuable to elaborate on future perspectives. 4. Authors should elaborate on links between vascular and inflammatory factors in the neurodegeneration.

Reply to reviewer: We greatly appreciate the enriching opinions. We read, and as far as possible, we include recommended bibliographies and revise. We rewrote several paragraphs and performed a major revision, as suggested by the reviewers, as noted in red hatching in the text.

We greatly appreciate the attention given to the analysis and contribution to the enrichment of this article.

Sincerely,

The authors

Round 2

Reviewer 1 Report

I congratulate the authors significantly improved their manuscript